# The Study of the Aorta Metallomics in the Context of Atherosclerosis

**DOI:** 10.3390/biom11070946

**Published:** 2021-06-25

**Authors:** Aleksandra Kuzan, Marta Wujczyk, Rafal J. Wiglusz

**Affiliations:** 1Department of Medical Biochemistry, Wroclaw Medical University, 50-368 Wroclaw, Poland; 2Institute of Low Temperature and Structure Research, Polish Academy of Sciences, 50-422 Wroclaw, Poland; m.wujczyk@intibs.pl (M.W.); r.wiglusz@intibs.pl (R.J.W.)

**Keywords:** ionomics, atherosclerosis, calcium, magnesium, iron, copper, chromium, zinc, manganese, cadmium, lead

## Abstract

Atherosclerosis is a multifactorial disease, for which the etiology is so complex that we are currently unable to prevent it and effectively lower the statistics on mortality from cardiovascular diseases. Parallel to modern analyses in molecular biology and biochemistry, we want to carry out analyses at the level of micro- and macroelements in order to discover the interdependencies between elements during atherogenesis. In this work, we used the Inductively Coupled Plasma Optical Emission Spectrometer (ICP-OES) to determine the content of calcium, magnesium, iron, copper, chromium, zinc, manganese, cadmium, lead, and zinc in the aorta sections of people who died a sudden death. We also estimated the content of metalloenzymes MMP-9, NOS-3, and SOD-2 using the immunohistochemical method. It was observed that with the age of the patient, the calcium content of the artery increased, while the content of copper and iron decreased. Very high correlations (correlation coefficient above 0.8) were observed for pairs of parameters in women: Mn–Ca, Fe–Cu, and Ca–Cd, and in men: Mn–Zn. The degree of atherosclerosis negatively correlated with magnesium and with cadmium. Chromium inhibited absorption of essential trace elements such as Cu and Fe due to its content being above the quantification threshold only if Cu and Fe were lower. Moreover, we discussed how to design research for the future in order to learn more about the pathomechanism of atherosclerosis and the effect of taking dietary supplements on the prevalence of cardiovascular diseases.

## 1. Introduction

Cardiovascular medicine seems to be developing towards surgery and molecular diagnostics [1,2], while basic research has not yet provided a complete answer to the question of the pathomechanism of atherosclerosis. The main functions of macro- and micronutrients are known and described, but the interdependence of the various elements remains unknown in many cases. Below, we describe the characteristics of each element that interested us in this project, emphasizing the context of its impact on the development of atherosclerosis.

Iron ion (Fe^2+^ and Fe^3+^) is the most abundant trace element in the human body [3]. Iron is required for hemoglobin and various enzymes, including the activity of nitric oxide synthase (NOS). However, its excess is very disadvantageous, because with its participation, the Fenton-type reaction occurs, whereby Fe^2+^ reacts with hydrogen peroxide to generate hydroxyl radicals and highly reactive intermediates, causing oxidative stress in the cell [4]. ROS generated in the presence of iron ions cause oxidation of LDL, which along with other factors, leads towards the development of atherosclerosis. The experimental data, however, do not agree on whether excess iron [5,6] or rather iron deficiency play a pivotal role in the development of cardiovascular diseases [7].

Copper ion (Cu^2+^) has a role in hemoglobin synthesis and immune function and is a cofactor for Cu/Zn superoxide dismutase. It is also crucial in the activity of other enzymes, such as lysyl oxidase, tyrosinase, ferroxidase, cytochrome c oxidase, and ceruloplasmin. Cu^2+^ ions are potentially as harmful as Fe^2+^ ions, due to the presence of the peroxide it generates a hydroxyl radical, in line with a Fenton-type reaction. Although, at the same time, as a component of SOD1 and SOD3, it has a very important antioxidant function. It is not known whether the pro-health properties of copper outweigh the negative potential of the element, because although for some high blood copper concentration is thought to be an independent risk factor for cardiovascular disease [8], others argue that copper deficiency is associated with atherosclerosis [9].

Calcium ion (Ca^2+^) is needed in the artery, especially for myocytes to contract, but it is also present in the tissue in the form of salts, mainly as hydroxyapatite, (Ca_10_(PO_4_)_6_(OH)_2_), and also as octocalcium phosphate (Ca_8_(PO_4_)·6.5H_2_O) and amorphous calcium phosphates (Ca_9_(PO_4_)_6_·nH_2_O), contributing to the stiffening of the arteries [10].

Magnesium ion (Mg^2+^) is the most abundant intracellular divalent cation and is a cofactor in more than 325 enzyme systems in cells [11]. Mg^2+^ also has a role in regulating vascular tone, thrombosis, vascular calcification, and proliferation and migration of endothelial and vascular myocytes [12].

Zinc ion (Zn^2+^) participates in more than 300 enzymatic systems and is a key player for various processes such as protein transcription control, glucose control, wound healing, digestion, and fertility. Zinc ion acts as an antioxidant and membrane stabilizer. In the context of atherosclerosis, it is important that Zn^2+^ is involved in the maintenance of immunity by regulating T-helper and suppressor cells, T-effector cells, and T-natural killer cells [13,14], and that it is an essential component of metalloproteases that degrade the extracellular matrix in the course of arterial remodeling during atherogenesis [15] and a cofactor of nitric oxide synthase (NOS)—an enzyme involved in blood pressure regulation with antiatherosclerotic function [14]. It is suggested that supplementing with zinc can reduce the risk of atherosclerosis and protect against myocardial infarction [16].

Manganese ion (Mn^2+^) is involved in the synthesis and activation of many enzymes (e.g., superoxide dismutase, arginase, glutamine synthetase, phosphoenolpyruvate decarboxylase, oxidoreductases, transferases, hydrolases, lyases, isomerases, and ligases), and thus is responsible for the regulation of glucose and lipid metabolism, influences the synthesis of proteins, vitamin C and vitamin B, and hormones, catalyzes hematopoiesis, and stimulates immune system processes [17].

Super oxide dismutase (SOD) is an enzyme from the group of oxidoreductases that catalyze the dismutation of the superoxide radical anion. It occurs in three isoforms: SOD1, the so-called cytoplasmic form, containing Cu and Zn; SOD2, the so-called mitochondrial form, containing manganese; and SOD3, the extracellular form, also containing Cu^2+^ and Zn^2+^ ions [18]. All SODs, including SOD2, are involved in anti-atherogenic mechanisms due to the inhibition of mitochondrial oxidative stress. In addition to preventing ROS production and DNA damage in mitochondria, SOD2 regulates endothelial myocytes’ proliferation and apoptosis, inhibiting the development of atherosclerosis [18]. Therefore, it is postulated that the determination of the manganese content in the vascular wall may be one of the prospective methods for the diagnosis of early stages of atherosclerosis, and also that Mn^2+^ ion supplementation could reduce endothelial dysfunction and lower cholesterol, preventing the development of atherosclerosis [17].

Chromium ion (Cr^3+^) in the third oxidation state is an important micronutrient necessary for the proper metabolism of carbohydrates; therefore, its deficiency is associated with diabetes. The mechanism of this dependency is that Cr^3+^ ion activates glucose transporter 4 and enhances insulin-stimulated glucose transport [19]. Diabetes mellitus is often a coexisting disease with atherosclerosis. Additionally, the relationship between chromium and atherosclerosis is also based on the effect of chromium on the adsorption of cholesterol. On cell cultures, it was shown that the presence of Cr^3+^ ion reduced the content of membrane cholesterol in adipocytes [19]. It is worth noting, however, that this element can also be present in the sixth oxidation state, being then associated with industrial exposure and toxicity [19,20].

Lead (Pb^2+^) and cadmium (Cd^2+^) ions are considered as micronutrients with no physiological function, and are cytotoxic [21]. It is reported that lead directly interrupts the activity of enzymes, competitively inhibits absorption of essential trace minerals, and deactivates antioxidant sulfhydryl pools [22]. Both elements induce the development of atherosclerosis and hypertension associated with oxidative stress [23,24]. The source of these elements is heavy industry and cigarette smoke—a single cigarette is estimated to contain 12 μg of Cd, and an average of 10% is inhaled during smoking [24].

## 2. Materials and Methods

### 2.1. Biological Material

The research material consisted of fragments of the human thoracic or abdominal aorta, which were part of the biological material taken for histopathological examination to determine the cause of death of the deceased. The material was collected during forensic autopsies carried out at the Department of Forensic Medicine at the Wroclaw Medical University. This work was carried out in accordance with the Declaration of Helsinki (2000) of the World Medical Association. This study was approved in terms of ethics by the Bioethics Committee of Wroclaw Medical University (no 577/2017).

The samples came from people who died a sudden death (range of 68 ± 15 years old); 10 samples were from women, 16 samples were from men. To allow the degree of atherosclerosis to be correlated with the content of selected elements, the six-degree scale according to the American Heart Association (AHA) scale [25,26] was used. The stages were defined as: I—early lesions, with single foam cells, II—fatty streaks, with intracellular lipid accumulation, III—pre-atheroma, where accumulation of foam cells and regions of extracellular lipids in the intima are characteristic, IV—atheroma, with the presence of a fatty core in which calcium deposits and/or cholesterol crystals may be present, V—fibroatheroma, where a fatty core with formation of a fibrous cap or formation of calcium deposits is characteristic, and the lesions reach the medial and/or adventitious layers, and VI—ruptured lesion, calcified lesion, or fibrotic lesion, where defects of the plaque surface, ulcers, hematomas, blood clots, significant reductions in vessel lumen, or large calcium deposits have been observed. For this study, aorta samples were randomly taken from thoracic and abdominal aorta territories. The distribution of samples according to the age of the patients and the degree of advancement of the aortic segment is presented in Figure 1.

### 2.2. ICP-OES Coupled Plasma Optical Emmision Spectrometer

Every specimen was thermally treated at 600 °C for 3 h in an electric furnace, and weighed before and after the thermal treatment. In order to fabricate an inorganic solution from the specimen residue, the remaining residue was transferred to a Teflon vessel containing 1 M ultrapure HNO_3_ (Sigma-Aldrich, Saint Louis, MI, USA). The vessel was placed in a microwave reactor for 60 min. In microwave-stimulated hydrothermal conditions, under autogenous pressure of 20 atm and at 250 °C, the solution was obtained. The procedure was repeated for every specimen. Resulting solutions were analyzed for the content of elements (Ca, Cd, Cr, Cu, Fe, Mg, Mn, Pb, Zn) using inductively coupled plasma optical emission spectrometry (ICP-OES). The total concentration of the trace elements was measured using an Agilent 5110 synchronous vertical dual view (SVDV) ICP-OES instrument equipped with an easy-fit quartz torch with standard 1.8 mm injector and a Seaspray nebulizer as a sample introduction system and a double-pass glass cyclonic spray chamber. The resulting solutions were measured versus simple standard solutions. The instrument was run under standard operating conditions, while combining radial and axial acquisition of radiation emitted by the vertical plasma over the entire wavelength range, in a single measurement.

### 2.3. Immunohistochemistry

The slides were dewaxed, rehydrated, and rinsed twice in phosphate-buffered saline. Antigen retrieval with HistoReveal (Abcam, Cambridge, UK) was performed as a pretreatment. Slides were incubated with 3% hydrogen peroxide for 10 min and then with Protein Block (Abcam, Cambridge, UK) for 30 min. Sections were incubated with anti-NOS-3 mouse monoclonal antibodies (sc-376751, 1:50, Santa Cruz, Dallas, TX, USA), anti-MnSOD mouse monoclonal antibodies (ab16956, 1:300, Abcam, Cambridge, UK), and anti-MMP-9 mouse antibodies (DuoSet Human MMP-9/TIMP-1 Complex, DY1449, R&D Systems, Minneapolis, MN, USA) at 4 °C for 18 h. The color reaction was performed using biotinylated goat anti-polyvalent, streptavidin peroxidase, and DAB substrate (ab64264, Abcam, Cambridge, UK). Nucleic and other basophilic proteins were counterstained with hematoxylin (ab220365, Abcam, Cambridge, UK). The negative control consisted of samples incubated without the primary antibody. Slides were dehydrated, closed with a glass coverslip using DPX (Avantor Performance Materials Poland S.A, Gliwice, Poland), and examined using an Olympus BX51 microscope.

### 2.4. Von Kossa Silver Test for Calcium

Von Kossa stain was used to visualize calcium deposits in histological tissue specimens. Paraffin preparations were dewaxed with xylene and rehydrated with a descending alcohol series (100–50%), then immersed with 5% silver nitrate (Sigma-Aldrich, Saint Louis, MI, USA) solution and exposed to bright sunlight for 20 min. The slides were washed in several changes of distilled water and un-reacted silver was removed with 5% sodium thiosulfate solution (Sigma-Aldrich, Saint Louis, MI, USA). As counterstain, the nuclei and cytoplasm were visualized with hematoxylin solution. Slides were dehydrated with an ascending alcohol series and xylene, closed with a glass coverslip using DPX (Avantor Performance Materials Poland S.A, Gliwice, Poland), and examined using an Olympus BX51 microscope.

### 2.5. Statistics

Statistical analysis was conducted using the data analysis software system Statistica 13.3 (TIBCO Software Inc., Palo Alto, CA, USA) software. Statistical significance level was set at *p* < 0.05. Outliers for all analyses except the preliminary analysis for mean analyte contents were removed. The differences in the values of the variables between the sexes and differences in the content of elements depending on the occurrence of Pb, Cr, and Cd were analyzed using the non-parametric Mann–Whitney U test. The correlation matrix was made using the Spearman correlation.

## 3. Results

### 3.1. Preliminary Analysis

#### 3.1.1. Average Analyte Contents

In the analyzed material, the mean content of calcium was 271 ± 169.5 mg/g of tissue; content of copper was 0.214 ± 0.222 mg/g of tissue; content of iron was 5.174 ± 9.428 mg/g of tissue; content of magnesium was 9.573 ± 6.363 mg/g of tissue; and content of zinc was 1.103 ± 0.91 mg/g of tissue. Four of the analytes had such low tissue contents that only some samples had detectable amounts. This was the case for cadmium, which had a detectable amount in eight cases; chromium, for which there were six samples above the assay’s quantification; manganese, for which 23 samples had values above quantification, and lead, which had detectable content in only four samples.

#### 3.1.2. Differences in the Values of Variables between the Sexes

Statistically significant differences in the median between the sexes occurred in the case of stage of atherosclerosis, age, Cu, Fe, Mg, Mn, and Zn, but in the case of Ca it was also close to being significant (*p* = 0.058). These data are presented in Figure 2.

Since the sex strongly differentiates practically all analyzed variables, further analyses were carried out in separate sexes.

### 3.2. Statistical Analysis of the Relationship between Parameters

#### 3.2.1. Correlation Matrix Including Ca, Cu, Fe, Mg, Mn, and Zn

In men, statistically significant correlations were observed between Cu and Ca, Mn and Ca, Mn and Cu, Zn and Cu, Zn and Fe, and Zn and Mn (*r* values in Table 1, scatter plots in Figure 3).

For women, similar correlations were observed between the following parameters related to Cu–Ca, Mn–Ca, Zn–Cu, and Zn–Fe pairs. Moreover, correlations between Fe and Cu and Zn and Ca were observed. The r values are in Table 2, the scatter plots in Figure 4.

#### 3.2.2. Analysis of Cd, Pb, and Cr Content

Only the data for women were used due to the larger sample size for this sex (there were more male patients, but there were more results above the detection limit for women).

In this case of investigation, two methods were proposed: (1) correlating (Spearman’s correlation) data for Cd, Cr, and Pb with other variables, and (2) stratifying the data according to Cr, Cd, and Pb concentrations above and below the detection limit (data were marked as “n” when below the detection limit and “y” when above the detection limit).

##### Correlations

There were significant correlations between Cd and Ca (negative), and Cu, Fe, and Zn (positive), as well as a negative correlation between Cd and the degree of atherosclerosis (Table 3).

##### Comparison of N vs. T

−Cd: no difference of any kind between n and y;−Cr: significantly higher values of Cu (*p* = 0.010) and Fe (*p* = 0.014) for n (i.e., below the Cr threshold);−Pb: significantly higher Cu values (*p* = 0.032) and almost significantly (*p* = 0.070) higher Fe for n (i.e., below the threshold Pb values).

#### 3.2.3. Analysis Elements vs. Degree of Development of Atherosclerosis

Such an analysis was performed for both women and men. No statistically significant relationship was observed for women. For men, there was a correlation between Ca and age, Cu and age, Fe and age, and Mg and degree of atherosclerosis. The r values are in Table 4, and the scatter plots in Figure 5.

#### 3.2.4. Correlation Analysis among Samples in Advanced Disease Stages and in Early Atheroma Development Stages

Another analysis was performed by dividing the samples into samples on insipient atheroma development stages (AHA grades I and II) and samples on advanced disease stages (AHA grades IV–VI). In both groups, a separate analysis of the correlations between the analytes was performed. In the group of samples in advanced disease stages, statistically significant negative correlations were found between Cd and Ca, and positive between Fe and Cd, Fe and Cu, Mg and Cr, Mn and Cu, Mn and Fe, Zn and Cu, Zn and Fe, and Zn and Mn (Table 5). In the samples in the early stages of atherosclerosis development, only positive correlations were found between Mg and Ca, Zn and Cu, and Zn and Mn (Table 6). Most of the correlations were consistent with those presented in Table 1, Table 2 and Table 3, but two were not observed in previous analyses—the correlation between Mg and Cr and between Mn and Fe in samples in advanced disease stages.

### 3.3. Analysis of Preparations with Immunohistochemical Detection of SOD2, MMP9, and NOS3 and Staining of Calcium Salts by the Von Kossa Method

A tendency was observed that the more advanced the atherosclerosis, the more SOD2 antigen was present in the artery wall. The antigen was clearly accumulated in the atherosclerotic plaque (Figure 6c,d), and smaller amounts of the antigen were also seen in the media (Figure 6e) and in the vasa vasorum (Figure 6f).

The occurrence of MMP9 was similar. Although the intensity of the reaction was generally much lower, the presence of the antigen was especially pronounced in plaques with highly developed atherosclerosis (Figure 7c) or in the layers below the plaque (Figure 7d,e).

NOS3 was generally found on the endothelium, both in slightly atherosclerotic aortas (Figure 8a) and in the vasa vasorum of the normal adventitia (Figure 8b). At high degrees of advancement of atherosclerosis, the endothelium was damaged and disintegrated, with no signs of NOS3 expression. Even though it is integral (Figure 8c), the presence of NOS3 was not observed over a developed atherosclerotic plaque.

Highly calcified preparations become crumbly and cannot be shown in a representative manner. The presence of calcium deposits is also visible macroscopically, and microscopically the concentration of calcium salts can be assessed by von Kossa staining. We present an example of a sample with low calcium salt content (Figure 9A(a,b)) and with relatively high calcium salt content (Figure 9B(c–e)).

## 4. Discussion

Many studies have examined the elemental content of serum [4,8,10,22,27,28], and there are very few analyses involving the artery itself. If they occur in the artery, they involve 1–2 elements [3,29]. We performed the analysis of as many as nine elements, hence the uniqueness and great value of our work.

Zhang recently (2020) prepared an extensive review paper in which he described ionomics in various disease states, including diabetes, cancer, and neurodegenerative disease, but of the cardiovascular diseases he only mentioned ischemic heart disease and thoracic aortic dissection, citing only four publications that presented data in this area (with only one having arteries as study material) [21]. There was no mention of atherosclerosis. Thus, one can see a gap in the literature.

It is difficult to compare our results with the results obtained in blood, because sometimes cardiovascular diseases are probably due to the fact that the level of the micronutrients decreases in the tissue and increases in the blood, or decreases in the blood, not necessarily being deficient in the tissues [4,24,30,31]. In the absence of other data, however, we will use comparisons to serum analyses.

### 4.1. Sex-Related Differences

Sex-related differences were found in many population studies examining the content of elements in biological samples; for example, in a study of Rambousková, Tubek, and Nawrot [27,32,33].

It is known that the content of some elements differs between the sexes, such as iron due to the compensatory leveling of the element after menstrual blood loss. In general, different effects of sex hormones on other endocrine systems (e.g., zinc metabolism is related to the renin–angiotensin–aldosterone system) and ion channels (cadmium ion has greater toxicity related to the influence of progesterone in calcium channels) are observed [32].

It is interesting that in some studies, some elements are tested almost exclusively for in one sex. For example, for chromium, only six population studies are really available describing the relationship between the level of chromium in patient samples (blood, serum, urine, toenail) and atherosclerosis, with three studies restricted to men only, and a fourth study population included 92% men [20].

Research was conducted on a group consisting of both women and men, and we analyzed them separately for the correctness of conclusions. Completely different types of research are necessary to decipher the causes of differences between the metabolism of these elements by sex.

### 4.2. Iron

According to our research, the amount of iron in the aorta decreases with age. We were unable to find any other work that would confirm our result by examining the content of this micronutrient in this tissue, but it is accepted in the literature on the subject that iron deficiency is prevalent in older age [34]. As older age is often associated with the severity of atherosclerosis, we also expected to discover the correlation that the more advanced the atherosclerosis, the less iron in the tissue. However, such an effect was not observed in our study. Other studies also do not give unequivocal answers as to whether iron is accumulated during atherosclerosis or whether its deficiency is associated with atherogenesis. Kali et al. reported that iron ion accumulation in arterial wall macrophages is increased in atherosclerotic lesions [35]. Tascic et al. also concluded that high iron levels may contribute to atherosclerosis [6]; similarly, Edvinsson stated that elevated body Fe ion stores are considered a risk factor for cardiovascular diseases [5]. The theory of the pro-atherogenic effect of iron ion is explained mainly by the influence of oxidative stress generated in the presence of Fe ion. In contrast, Lam et al. and others postulate that iron deficiency plays an essential role in the development of cardiovascular diseases [7], especially heart failure [36,37]. The proposed mechanism is that the chronic increase in cardiac output caused by hypoxia as a result of iron deficiency may lead to arterial remodeling of central elastic arteries such as the aorta, and this in turn results in arterial enlargement and compensatory arterial intima–media thickening, and arteriosclerosis [38]. Moreover, Li et al. stated that low levels of circulating and aortic iron were associated with vascular myocyte dysfunction and aortic instability [3], but this was a study strictly aimed at the formation of aneurysms. Due to the lack of unequivocal results of clinical trials, which would lead to a consensus, based on our own and literature studies, it could be concluded that iron level is not a parameter that unequivocally predisposes a patient to the development of atherosclerosis.

Statistically significant positive correlations were shown between Fe and Cu in women and Fe and Zn ions in both sexes. Stadl et al. also found a positive Fe–Zn ion pair correlation in arteries [39], and at least two other publications showing the results of the correlation of these parameters in serum were consistent with our results: Tasic et al. reported a positive significant correlation for Cu and Fe in the serum of patients after carotid endarterectomy [6] and Ari reported a positive correlation between serum Fe and Zn of hemodialysis patients [8]. A question arises as to the reason for such a high correlation coefficient (0.9−0.6) between these microelements. We propose a theory that the more iron, the more copper and zinc accumulate in Cu/Zn SOD by a compensation mechanism. To verify this hypothesis, SOD1 and SOD3 in the aortas should be determined with a quantitative test and compared with the iron content. This is a future direction for us.

To demonstrate the presence of iron-containing metalloenzyme as a cofactor in the arterial wall, we chose endothelial nitric oxide synthase (eNOS), also known as nitric oxide synthase-3 (NOS-3), and performed a series of immunohistochemical reactions with the anti-NOS3 antibody. The enzyme plays crucial role in maintaining the proper functions of the vascular endothelium, counteracting hypertension, and the initiation of atherosclerosis. In the case of hypertension, hypercholesterolemia, smoking and diabetes, and oxidative stress, the NOS3 cofactor is oxidized, which results in down regulation of aortic nitric oxide and antioxidant systems [40,41]. We also observe a decrease in the number of NOS3-expressing endothelial cells in samples with highly advanced atherosclerosis.

### 4.3. Calcium and Magnesium

Calcium in the aorta can reach a dozen or even several dozen percent (in terms of dry weight), according to our research or the research of Joh et al. [42]. It was found that in the carotid arteries, the calcium content may be even higher than in the aorta [42]. Calcium overload can have serious clinical consequences [42].

Our research hypothesis was that the calcium content would increase with the severity of atherosclerosis. The premise here was the fact that calcification is often associated with atherosclerosis; micro-calcification is even named a hallmark of atherosclerosis [43]. Meanwhile, the experimental results do not confirm our hypothesis, because calcium correlates with age rather than with the severity of the disease. Wróbel et al. also found such a correlation by examining aortic samples [44]. On the other hand, here, a negative correlation between the degree of atherosclerosis and the content of magnesium in the artery was also observed. It could also be related to the relationship with calcification because of the known antagonistic effects of magnesium on calcium functions, including hydroxyapatite formation and calcium transport into cells. It is proved that serum magnesium concentration is inversely correlated with vascular calcification [10]. The mechanism of this dependency is based on the fact that Ca^2^ and Mg^2+^ ions have an antagonistic effect on hydrolysis of pyrophosphate (PPi) in the aortic wall. Magnesium ions inhibit PPi hydrolysis and calcium promotes it. PPi is the main endogenous inhibitor of vascular calcification [10].

The observation of a negative correlation between Mg content and atherosclerosis was noted by Tzanakis et al., who analyzed a group of patients with chronic renal failure [45]. Of course, the data were based on serum and did not directly determine the severity of atherosclerosis, only the thickness of the internal membrane of the middle carotid artery, but the conclusion was consistent with ours—the less magnesium in the tissue/serum, the more advanced atherosclerosis. The effect of magnesium ion serum deficiency in hemodialysis patients on cardiovascular mortality was also studied, and there appeared to be a significant association [46]. A meta-analysis of Leenders et al. supported this conclusion [47]. It is worthwhile, in our opinion, to extend the study to the entire population, not only to patients with kidney disease, because from our results we can conclude that the relationship between magnesium content and atherosclerosis may apply to everyone.

This raises the question of the value of magnesium supplementation in people at higher risk of cardiovascular disease. While some studies support the value of magnesium supplementation in these individuals, some studies have reported inconsistent benefits and raised potential adverse effects of magnesium overload. As such, there is currently no firm recommendation for routine magnesium supplementation except when hypomagnesemia has been proven [12]. Rather, it is recommended to consume a healthy diet (rich in green leafy vegetables, cereal, nuts, and legumes) that provides the recommended amount of Mg [11].

### 4.4. Manganese

The expected effect was to observe a greater expression of SOD2 in atherosclerotic plaques than in control aortic fragments without signs of atherosclerosis. We can confirm this observation for most of the analyzed cases. Knowing that SOD2 is a manganese-containing enzyme, we expected the result that the manganese content correlated with the degree of advancement of atherosclerosis, especially since there are publications that describe how the increase in Mn concentration in blood is associated with atherosclerosis [4,27]. However, such an effect was not observed in the studied research material. The reason may be related to the fact that this group was too small or that manganese was present in many enzymes other than SOD2, and not all these proteins had anti-atherosclerotic effects, so the relationship we expected may not be a straight line. The lack of this relationship was demonstrated earlier, e.g., Mendis (1989) stated that the difference between the Mn contents of normal and atherosclerotic aortic tissue was not significant [48].

A negative correlation in both sexes was shown between the manganese content and calcium. This is an expected result since it is known that Ca ion is an antagonist of Mn ion and both enter the cell through Ca ion channels [49]. It was shown that high intakes of calcium impair manganese absorption [50].

Moreover, statistically significant positive correlations were also observed for Mn–Zn and Mn–Cu pairs in men, with a very high correlation coefficient (0.84 for Mn–Zn pair). The lack of more data makes it impossible to discover why this result is not also observed in women. Positive Mn–Zn correlations were also observed in serum by Ari [8] and Mehra et al. in the hair [51]. In people with advanced atherosclerosis, we also observed a positive correlation for the Mn–Fe pair, as did others [8].

We did not find a report showing a positive relationship between manganese and copper. We expected this effect because since zinc correlates positively with copper and both are involved in enzymatic mechanisms of protection against oxidative stress, the two elements would be in a similar relationship with manganese, which is also involved in the same functions; for example, in the context of superoxide dismutase.

### 4.5. Copper

Ageing is an important risk factor for atherosclerosis [9]. Relying on literature data which report that Cu deficiency leads to increased LDL and triglycerides and decreased HDL, increased susceptibility of lipoproteins and tissues to oxidation, increased blood pressure, and increased atherosclerosis [9,52], the expected result in our study was therefore to observe a negative correlation between Cu and age and a positive correlation between Cu and disease severity. We were able to observe the first correlation, so we confirm that the older the patient, the lower the copper content in the vessels. We have no basis to confirm that copper levels may increase in people with atherosclerosis [27] or to confirm that Cu may be a risk factor for cardiovascular disease, as proposed by Ari et al. [8]. Moreover, Ari’s results are very different from ours, as the correlations he obtained for Mn–Cu, Zn–Cu, and Fe–Cu pairs had the opposite sign. The simplest explanation is that since that team analyzed serum of patients and we analyzed artery fragments, perhaps the content of copper in serum is not proportional to the content of copper in tissues. This would need to be verified, however, because there are currently no data available to resolve this issue.

It is worth mentioning that the study of Lee et al. and ours showed a positive correlation between Zn and Cu [30]. This was also a study on sera, in addition, in critically ill patients, so these results should be compared very carefully, but they show that copper and zinc were interdependent. There was also a review paper on the Zn–Cu pair’s relationship, which detailed the possible diagnostic use of Zn and Cu levels in the context of mortality in the elderly [53].

Iskra found a positive Cu–Ca correlation in sections of the arteries [54]. We observed quite a high and a negative correlation (*r* = −0.7). The correctness of our result may be proved by the results of Tasić et al., who, admittedly, analyzed only Cu and Zn, but also assessed the degree of plaque calcification, and it turns out that in highly-calcified plaques, the level of copper was lower [29]. We have no basis for recognizing the molecular basis, but we may suspect some kind of competition between Ca and Cu ions.

### 4.6. Zinc

Although we unequivocally observed that the more advanced the atherosclerosis, the more MMP9, we did not observe that zinc, which is a key element of this and other metalloproteinases, correlated with the severity of atherosclerosis. There is also a theory that it is rather zinc deficit that is associated with atherosclerosis [14], and zinc itself promotes lesion stability by binding to matrix components [39]. It would be good to verify this theory by conducting research on a large group of patients by analyzing paired samples—sera and arteries collected during cardiovascular surgical interventions. It is probable that a zinc compound grows in the arteries and decreases in the serum during atherogenesis.

The zinc content appears to correlate with the content of many micronutrients. We have already discussed the Zn–Cu, Zn–Fe, and Zn–Mn pair correlations above. Now, we are left to discuss the Zn–Ca pair’s correlation. Imaging studies show that zinc and calcium ions co-localize in areas of atherosclerotic plaque mineralization [55], but their quantitative relationship turned out not to be positive; quite the opposite. In the case of women, was observed that the correlation was statistically significant, negative, and quite high (*r* = −0.75).

Zn ion is considered a calcium channel blocker. It was reported that zinc ion increases intracellular accumulation of Ca ions, resulting in stiffened arteries, but its deficiency could reduce vasodilatation by signal transmission disturbances at the receptor level [22].

### 4.7. Cadmium

Most publications agree that the presence of cadmium in tissues is pro-atherogenic. High blood levels of Cd are reported to be associated with the initiation of atherosclerosis, which may be due to the hypertensive effects of Cd and its ability to cause endothelial cell damage via increasing oxidative stress [4,24,56]. 

Confusing data has been received, because in our study, atherosclerosis correlated negatively with cadmium. Although it seems illogical, we are not the first to observe the inhibitory effect of cadmium on atherosclerosis. Meijer et al. conducted experiments on rabbits with induced atherosclerosis through a high-cholesterol diet, and it turned out that cadmium supplements in food decrease atherosclerosis in a dose-related fashion [57].

We do not exclude that our result could be accidental, and if the study population were larger the result could differ, but we also consider that cadmium may indeed influence cholesterol metabolism, inhibiting the progress of atherogenesis.

The series of statistically significant correlations obtained with cadmium (Cd–Ca, Cd–Cu, Cd–Fe, and Zn–Cd pairs) is confirmed in some publications, and in others the correlation results are quite the opposite. For example, Ari, when analyzing the sera of patients, like us, observed a positive correlation between Cd and Cu, but the correlation between Cd and Fe and Cd and Zn in his study had the opposite sign [8]. Szlacheta, like us, described a negative correlation between Cd and Ca in the serum of heavy industry workers [58], but Nawrot et al., based on urine tests from a similar population, showed a correlation with the reverse sign [33]. It should be emphasized once again that comparison tests that are not performed on the arteries and on the serum must be very careful; therefore, the differences that appear do not discredit the correctness of our results, but it is necessary to confirm these data on the basis of a larger study.

### 4.8. Chrome and Lead

Typing “chrome artery atherosclerosis” into PubMed gives practically zero results on chromium content in the arteries. Hence, it is another case where we have no way of comparing our result to other studies. It was found that significantly higher values of Cu and Fe were present in samples with Cr content below the threshold values. Where the chromium content in the tissues is undetectable, the amount of Cu and Fe is much higher. Therefore, it has been suspected that chromium competes with Cu and Fe for a binding site in metal-binding proteins such as albumin and transferrin, and in the case of arteries, ferritin may be such a protein. The rationale behind this could be that an excess of chromium that binds to transferrin may cause anemia [59].

The analysis of samples with highly advanced atherosclerosis revealed a positive correlation between the content of chromium and magnesium. No other publication was found reporting such a dependence. However, it can be concluded that the synergistic relationship between chromium and magnesium indeed is based on the results of Whitfield et al., who described that supplementation with chromium and magnesium promotes the reduction in cholesterol and triglyceride levels [60].

As was noted in the introduction, chromium (III) ion deficit may influence the arteriosclerotic process via lack of blood glucose regulation [22]. That is why many diabetic and atherosclerotic patients take supplements that contain chromium ion [20]. Our research has shown that overuse of chromium can decrease the content of copper and iron in the tissues. Other authors also have found arguments that chromium supplementation is controversial [20].

The situation is very similar for lead as for chromium. Here, it was also found that in samples with lead content of zero or close to zero, the content of copper and iron was significantly higher. It probably inhibits absorption of essential trace elements such as Cu and Fe [22]. Perhaps it is competing with Fe and Cu for a site in ferritin, metallothionein, or some other protein. There is no unequivocal answer as to the mechanism, but it is unambiguous that there is a relationship between anemia from iron deficiency and elevated blood lead concentrations [61]. It has been not shown that there is a relationship between the degree of advancement of atherosclerosis and the content of lead in the tissues, but it should be emphasized that only five patients from the analyzed group had detectable levels of lead. It is likely that the multiplication of the test group would allow us to observe the relationship of Pb content with the progression of atherosclerosis.

A very clear limitation of our project is the lack of information regarding whether the donor of the aorta was a smoker, or other clinical data. Unfortunately, the forensic physician, who took the sample, did not have access to the history of the disease, nor, of course, was able to make an interview. It is expected that smokers could demonstrate elevated levels of cadmium and lead. Navas-Acien reported that cadmium and lead were both elevated in the serum of smokers [62].

## 5. Conclusions

One of our most important findings was that arterial calcium content increases with age and copper and iron level decline. The content of magnesium and cadmium negatively correlated with the severity of atherosclerosis. Many relationships between individual micronutrients were also described, many of them having never been reported in the literature on the subject before.

Practical conclusions that can be drawn based on this and similar studies concern supplementation. It is worth enriching the diet to protect against cardiovascular diseases. The pharmaceutical industry proposes iron, zinc, magnesium, and chromium (III) ions, influencing the well-being of arteries and glucose–lipid metabolism. It can be confirmed that magnesium deficiency might predispose individuals to the development of atherosclerosis. It should be remembered that each element is necessary only in a certain concentration, and too much can generate more harm than good, e.g., copper or iron, the presence of which promotes the formation of reactive oxygen species. Hence, attention has been paid to the need for great caution in reaching for supplements, but also in interpreting the data on the topic, because there is still not enough research to broadly and comprehensively analyze the relationships between the structural and dynamic elements of arteries related to the cause-and-effect mechanisms of atherosclerosis.

## Figures and Tables

**Figure 1 biomolecules-11-00946-f001:**
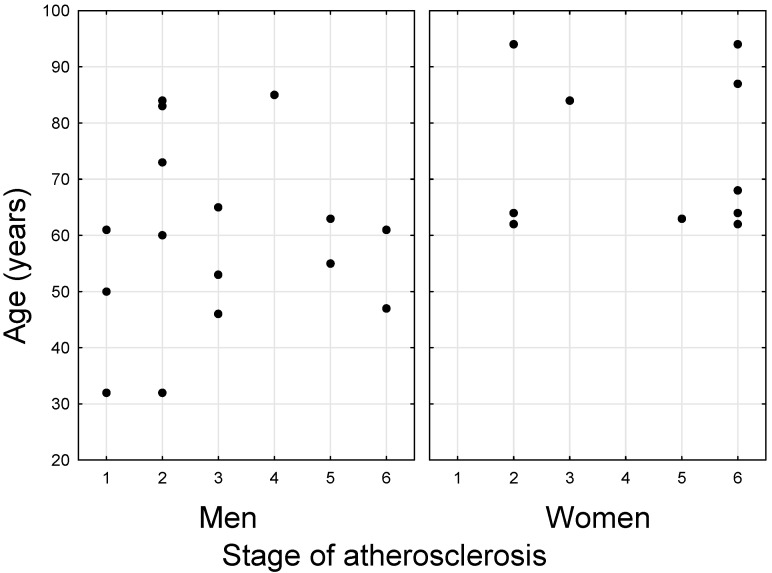
Sample distribution depending on the age of the donor and the severity of atherosclerosis in the aortic sample.

**Figure 2 biomolecules-11-00946-f002:**
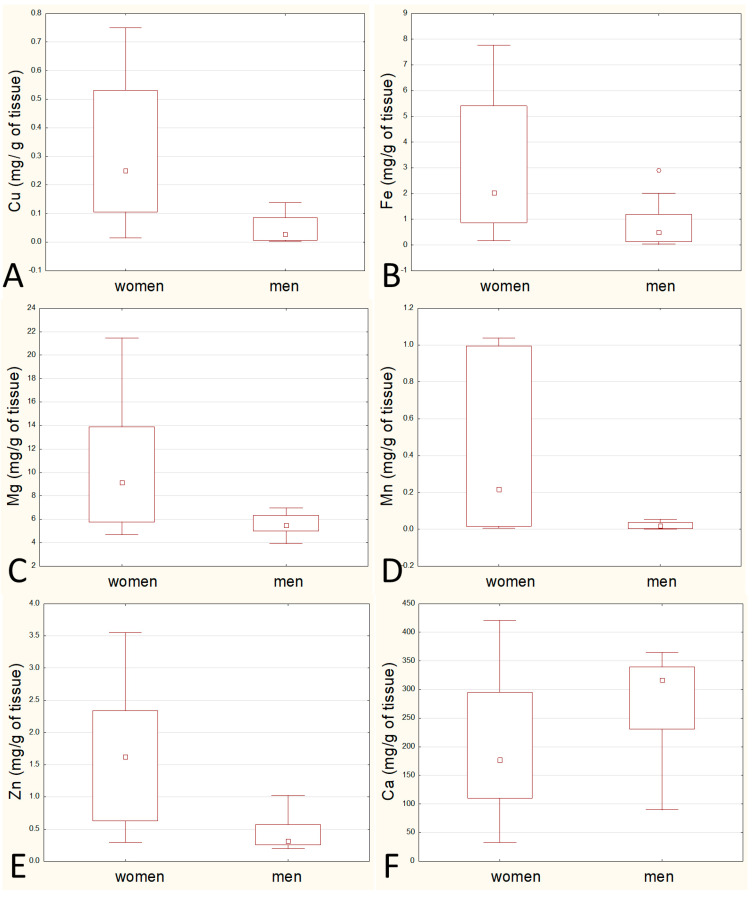
Comparison of the content of elements between the sexes: comparison of the content of copper (**A**), iron (**B**), magnesium (**C**), manganese (**D**), zinc (**E**) and calcium (**F**).

**Figure 3 biomolecules-11-00946-f003:**
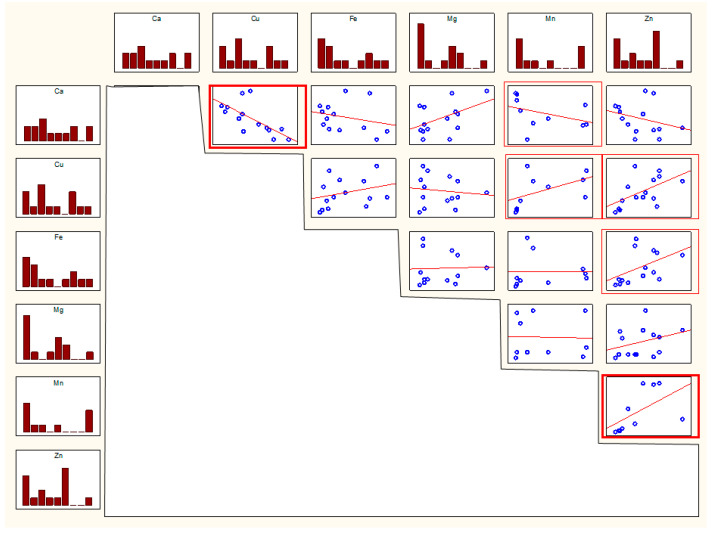
Scatterplot for pairs of elemental parameters in men. Thick frame—*p* < 0.01, thin frame—*p* < 0.05.

**Figure 4 biomolecules-11-00946-f004:**
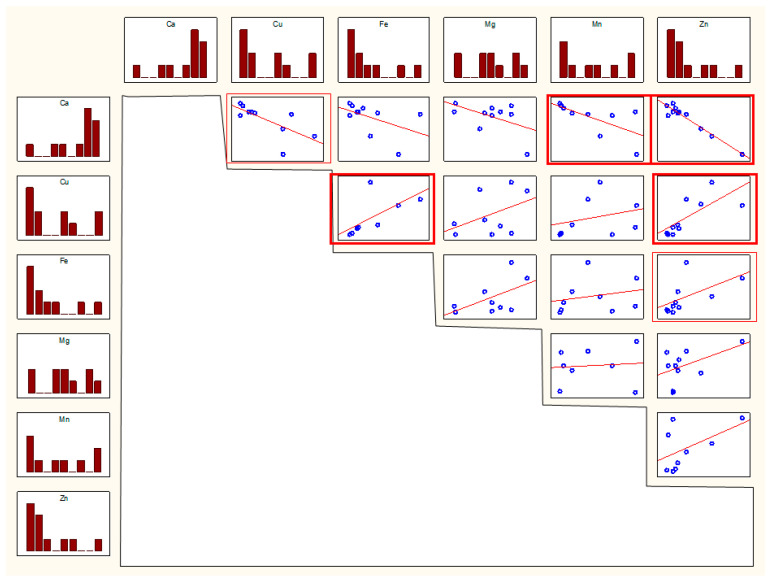
Scatterplot for pairs of elemental parameters in women. Thick frame—*p* < 0.01, thin frame—*p* < 0.05.

**Figure 5 biomolecules-11-00946-f005:**
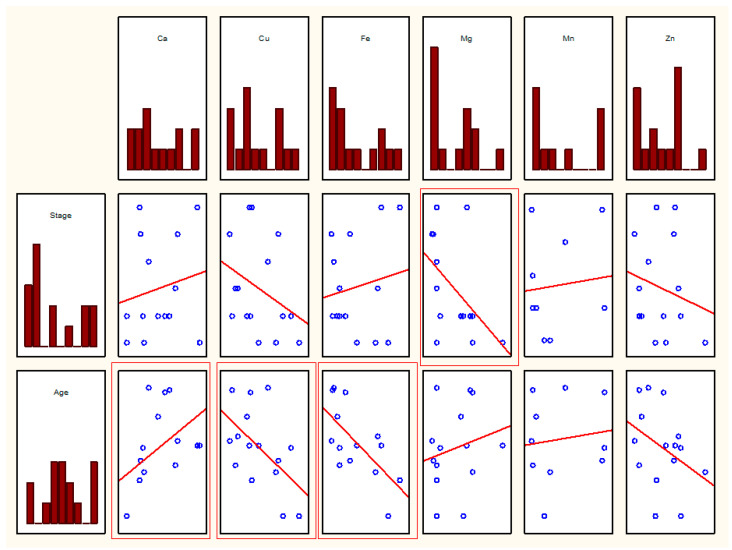
Scatterplot for pairs of elemental parameters considering age and stage of atherosclerosis in men. Thin frame—*p* < 0.05.

**Figure 6 biomolecules-11-00946-f006:**
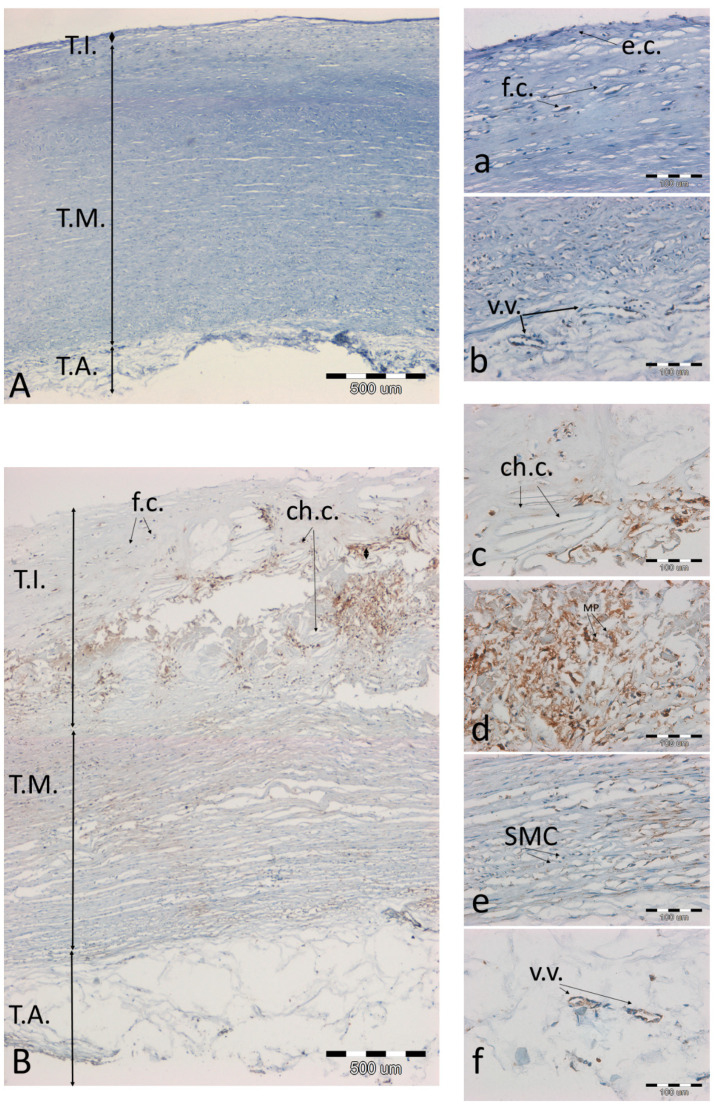
Immunodetection of the SOD2 antigen in a representative artery with a low degree of atherosclerosis (**A**(**a**,**b**) and high degree of atherosclerosis (**B**(**c**–**f**)). (**A**,**B**)—the entire aortic section, 40× magnification, scale bar 500 µm; (**a**–**f**)—magnification 200×, scale bar 100 µm. Abbreviations in figures: T.A.—tunica adventitia; T.I.—tunica intima; T.M.—tunica media; ch.c.—cholesterol clefts; v.v.—vasa vasorum; f.c.—foam cells; SMC—smooth muscle cells; e.c.—endothelial cells; a.—adipocytes; ECM—extracellular matrix; MP—macrophages.

**Figure 7 biomolecules-11-00946-f007:**
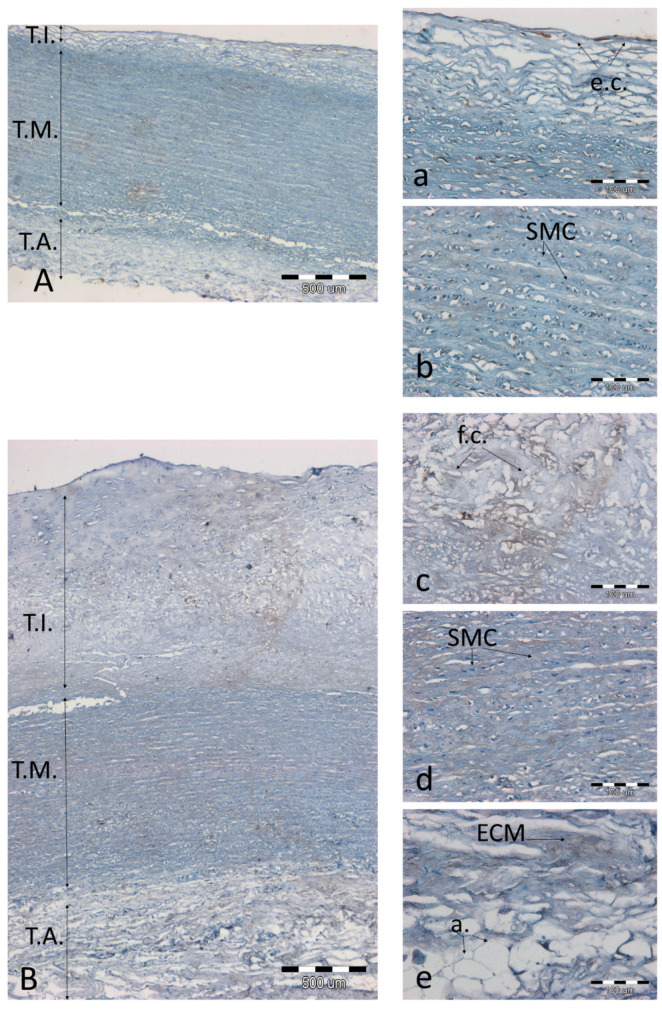
Immunodetection of the MMP-9 antigen in a representative artery with a low degree of atherosclerosis (**A**(**a**,**b**)) and high degree of atherosclerosis (**B**(**c**–**e**)). (**A**,**B**)—the entire aortic section, 40× magnification, scale bar 500 µm; (**a**–**e**)—magnification 200×, scale bar 100 µm. Abbreviations in figures: T.A.—tunica adventitia; T.I.—tunica intima; T.M.—tunica media; ch.c.—cholesterol clefts; v.v.—vasa vasorum; f.c.—foam cells; SMC—smooth muscle cells; e.c.—endothelial cells; a.—adipocytes; ECM—extracellular matrix; MP—macrophages.

**Figure 8 biomolecules-11-00946-f008:**
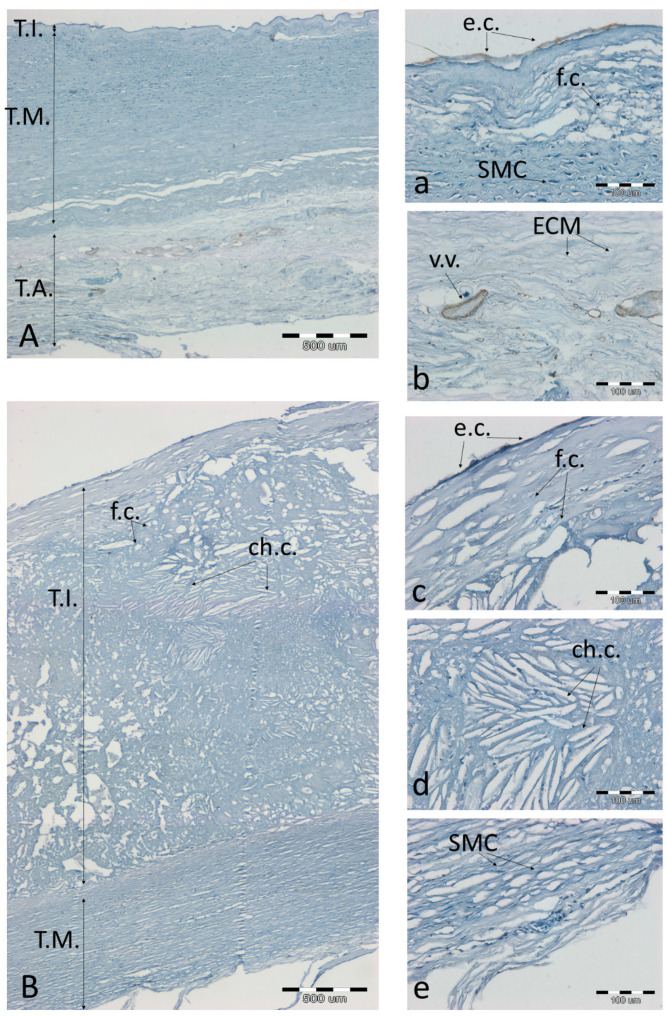
Immunodetection of the NOS-3 antigen in a representative artery with a low degree of atherosclerosis (**A**(**a**,**b**)) and high degree of atherosclerosis (**B**(**c**–**e**)). (**A**,**B**)—the entire aortic section, 40× magnification, scale bar 500 µm; (**a**–**e**)—magnification 200×, scale bar 100 µm. Abbreviations in figures: T.A.—tunica adventitia; T.I.—tunica intima; T.M.—tunica media; ch.c.—cholesterol clefts; v.v.—vasa vasorum; f.c.—foam cells; SMC—smooth muscle cells; e.c.—endothelial cells; a.—adipocytes; ECM—extracellular matrix; MP—macrophages.

**Figure 9 biomolecules-11-00946-f009:**
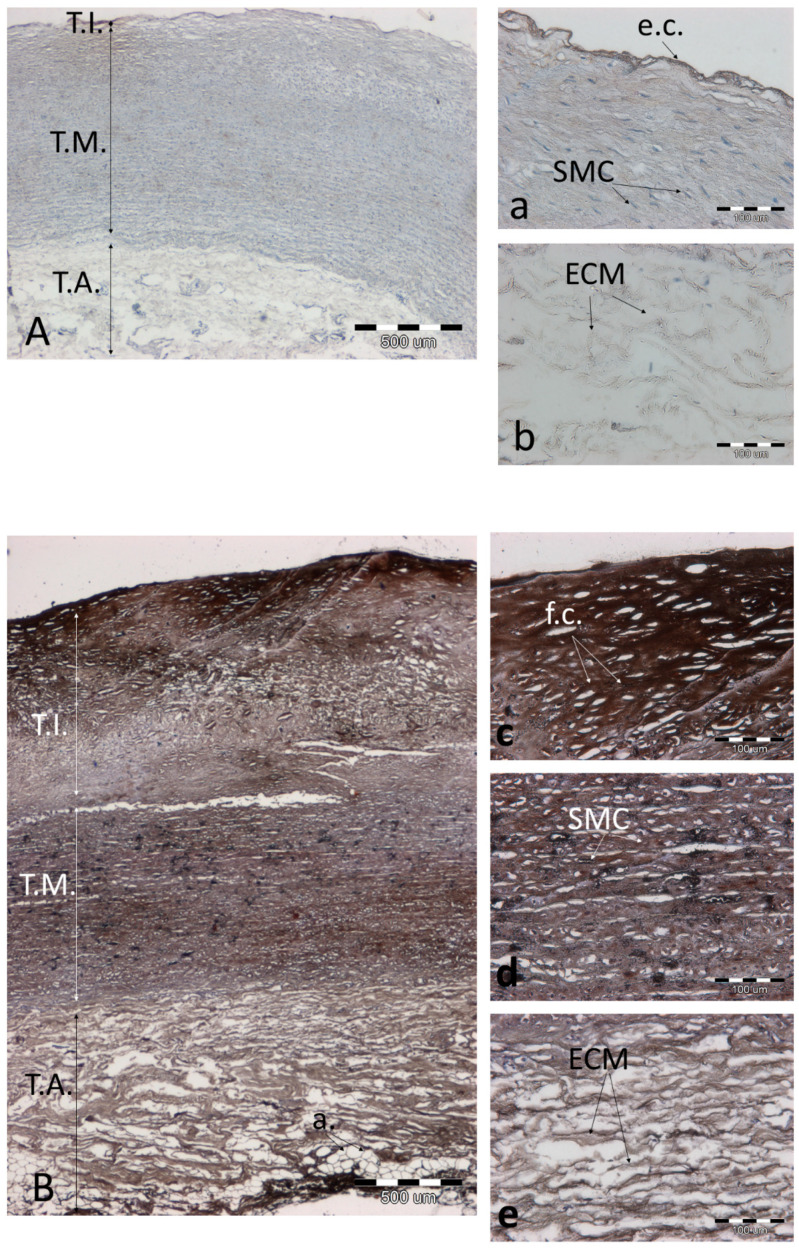
The effect of von Kossa staining in a representative aorta at a low level of atherosclerosis (**A**(**a**,**b**)) and an aorta at a high degree of atherosclerosis (**B**(**c**–**e**)). (**A**,**B**)—the entire aortic section, 40× magnification, scale bar 500 µm; (**a**–**e**)—magnification 200×, scale bar 100 µm. Abbreviations in figures: T.A.—tunica adventitia; T.I.—tunica intima; T.M.—tunica media; ch.c.—cholesterol clefts; v.v.—vasa vasorum; f.c.—foam cells; SMC—smooth muscle cells; e.c.—endothelial cells; a.—adipocytes; ECM—extracellular matrix; MP—macrophages.

**Table 1 biomolecules-11-00946-t001:** The values of the correlation coefficients for the content of analytes in the tissue of men. Statistically significant values are marked with an asterisk.

	Cu	Fe	Mg	Mn	Zn
Ca	−0.74 *	−0.39	0.36	−0.65 *	−0.33
Cu		0.47	−0.01	0.72 *	0.61 *
Fe			0.09	0.52	0.60 *
Mg				0.31	0.42
Mn					0.84 *

**Table 2 biomolecules-11-00946-t002:** The values of the correlation coefficients for the content of analytes in the tissue of women. Statistically significant values are marked with an asterisk.

	Cu	Fe	Mg	Mn	Zn
Ca	−0.70 *	−0.6	−0.38	−0.82 *	−0.75 *
Cu		0.90 *	0.33	0.38	0.78 *
Fe			0.43	0.4	0.76 *
Mg				0.29	0.31
Mn					0.37

**Table 3 biomolecules-11-00946-t003:** The values of the correlation coefficients for the content of analytes in the tissue of women, considering Cd, Cr, and Pb. Statistically significant values are marked with an asterisk.

	Ca	Cu	Fe	Mg	Mn	Zn	Stage of Atherosclerosis	Age
Cd	−0.94 *	0.83 *	0.83 *	0.70	0.70	0.83 *	−0.83 *	−0.68
Cr	−0.40	1	1	0.80	0.50	0.20	−0.26	−0.33
Pb	0.20	0.50	0.60	0.40	−0.50	1		−0.63

**Table 4 biomolecules-11-00946-t004:** The values of the correlation coefficients for the content of analytes in the tissue of men considering age and stage of atherosclerosis. Statistically significant values are marked with an asterisk.

	Ca	Cu	Fe	Mg	Mn	Zn
Stage	0.12	−0.37	0.01	−0.58 *	−0.17	−0.23
Age	0.61 *	−0.55 *	−0.65 *	0.23	0.07	−0.36

**Table 5 biomolecules-11-00946-t005:** The values of the correlation coefficients for the content of analytes in samples in advanced atherosclerosis stages. Statistically significant values are marked with an asterisk.

	Cd	Cr	Cu	Fe	Mg	Mn	Pb	Zn
Ca	−0.90 *	0.3	−0.44	−0.37	0.11	−0.37	−0.4	−0.4
Cd			1	0.90 *	0.3	0.8		0.7
Cr			1	1	0.90 *	0.8	0.5	0.6
Cu				0.73 *	0.22	0.65 *	0.6	0.75 *
Fe					0.29	0.65 *	0.8	0.77 *
Mg						0.27	0.6	0.31
Mn							−0.5	0.67 *
Pb								1

**Table 6 biomolecules-11-00946-t006:** The values of the correlation coefficients for the content of analytes in samples in early atheroma development stages. Statistically significant values are marked with an asterisk.

	Cu	Fe	Mg	Mn	Zn
Ca	−0.52	−0.27	0.81 *	0.33	−0.21
Cu		0.52	−0.32	0.59	0.71 *
Fe			0.17	0.61	0.59
Mg				0.44	0.18
Mn					0.78 *

## Data Availability

Data are available from the authors upon request.

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
