# Peer review of "The Study of the Aorta Metallomics in the Context of Atherosclerosis"

_biomolecules, 2021, doi:10.3390/biom11070946_

Round 1

Reviewer 1 Report

The paper covers a topic that it is not new: how redox balance and elements are implicated in the dramatic changes of artery walls and the severity of atherosclerosis in different territories.

This does not diminish the scientific relevance of the topic. The same questions remain valid and largely unanswered.

In more recent years this approach has been revisited in multiple contexts: influence of atmospheric pollution in the development of atherosclerosis related diseases, dietary supplements, gender and life style association with severity of atherosclerosis in different territories, variability in the morphology and biological characteristics of arteries in different territories (aorta, carotid, brachial, femoral, coronaries, etc.).

Thus, the study carried out is potentially interesting, specially due to the focus on the artery wall. However, the paper is merely descriptive for a poorly defined health condition and for a group of individuals which were not properly characterized. Therefore, reported elemental contents and immunohistochemical data for MMP-9, SOD, NOS are merely descriptive becoming trivial as their combined associations were not done having into account lesion types or other relevant characterization of atheroma severity. Consequently, discussion became speculative. Apart from the disregarded features mentioned above, the paper has other major inconsistencies.

Major concerns and study limitations:

  1. Sample characterization is missing according to the AHA lesion classification
  2. Group characterization – how samples were distributed by lesion type and gender.
  3. “Sudden death” is not defined. It can be anything from an accident to a disease condition. Criteria used for sample selection is unknown.
  4. How samples from thoracic and abdominal aorta were considered for analysis (elemental concentrations and immunohistochem) – i) two aorta regions analysed for each individual; ii) a piece of each region taken from each individual and pooled; iii) random selection.
  5. Which samples and how many of each group were analysed for both elemental concentrations and immunohistochemistry
  6. Elemental concentration values reported for artery vessels are extremely high, assuming that they are expressed in dry weight (dw) basis: 500-8000 mg/kg for Fe, 250-3500 mg/kg for Zn, 100-1000 mg/kg for Mn, 50-700 mg/kg for Cu, etc..

Ca concentrations reported in the paper are comparable to bone 17.5 – 33%! Even in calcified human arteries the concentration values for Ca rich regions in plaques, rarely exceed 5-6% (dw basis) (ref 1).

For other elements, such as trace elements Fe and Zn, their concentrations in healthy-normal tissues, including arteries are of the order of mg/kg dw: Fe, 90-400 mg/kg (except liver and lung where Fe may reach higher levels) and Zn 40-300 mg/kg (ref 1-3). That is the reason they are called trace elements – concentrations of 10-6 and below. Average Fe and Zn contents do not seem to differ too much between non-involved tissues of the artery wall (excluding elastic lamina). However, in the atheroma Ca, Fe and Zn concentration can drastically change. In atherosclerotic human coronaries Fe and Zn decrease whereas Fe may be increased in aorta (ref 1).  Even in rabbit tissues the elemental concentrations are similar to those detected in humans (ref 4).

7. Concentration values below detection limit were considered 0 (zero). If the value is below detection limit is a missing value in statistical terms; it cannot be assumed to be zero. Is completely wrong to establish median or average values considering missing values as 0 (zero) concentration.

8. Frequency of lesion types per group is not known so any relationship established with the variable “stage of atherosclerosis” is useless. The histograms in Figures with correlations may be related to lesion type frequency classes but it is very difficult to depict anything when the population characteristics is unknown.

9. Figures 5 to 8 - The immunohistochemical images are difficult to assess. No indications of artery wall stratification/tissue details, changes due to atheroma progression, etc..

In conclusion, authors lost the opportunity to make a relevant comparison between location of the specific proteins and lesion type with elemental concentrations. To achieve this objective, the study design should be clear and consistent. All the variables (lesion type, demographics, immunohistochem, elements) should exist for each individual. Also revising elemental concentration values produced and providing evidence of quality control for ICP-OES data is mandatory.

References

  1. Pinheiro T, et al. Nuclear microprobe applied to the study of coronary artery walls - A distinct look at atherogenesis. Cellular and Molecular Biology. 1996;42:89-102. PMID: 8833670. (complementary data in: Pallon J et al. A view on elemental distribution alterations of coronary artery walls in. Nuclear Instruments & Methods in Physics Research B. 1995;104:344-350. DOI:10.1016/0168-583X(95)00453-X.)
  2. Pinheiro T, et al. Elemental distribution in the human respiratory system and excretion organs: Absorption and accumulation. X-Ray Spectrometry. 1997;26:217-222. DOI:10.1002/(SICI)1097-4539(199707)26:4<217::AID-XRS215>3.0.CO;2-W.
  3. Lopes PA, et al. Systemic markers of the redox balance and apolipoprotein E polymorphism in atherosclerosis - The relevance for an integrated study. Biological Trace Element Research. 2006;112:57-75. DOI:10.1385/BTER:112:1:57.
  4. Minquin R, et al. Correlation of iron and zinc levels with lesion depth in newly formed atherosclerotic lesions. Free Radical Biology & Medicine, Vol. 34, No. 6, pp. 746–752, 2003. doi:10.1016/S0891-5849(02)01427-2 (complementary studies: Free Radical Biology & Medicine 38 (2005) 1206– 1211. doi:10.1016/j.freeradbiomed.2005.01.008; Free Radical Biology & Medicine 41 (2006) 222-225 doi:10.1016/j.freeradbiomed.2006.03.017)

Author Response

We would like to thank the Reviewer for his commitment and time spent analyzing our manscript. Thank you for the opportunity to revise the manuscript and for all valuable tips. We tried to respond precisely to each remark. Below (in green) our answers.

The paper covers a topic that it is not new: how redox balance and elements are implicated in the dramatic changes of artery walls and the severity of atherosclerosis in different territories.

This does not diminish the scientific relevance of the topic. The same questions remain valid and largely unanswered.

In more recent years this approach has been revisited in multiple contexts: influence of atmospheric pollution in the development of atherosclerosis related diseases, dietary supplements, gender and life style association with severity of atherosclerosis in different territories, variability in the morphology and biological characteristics of arteries in different territories (aorta, carotid, brachial, femoral, coronaries, etc.).

Thus, the study carried out is potentially interesting, specially due to the focus on the artery wall. However, the paper is merely descriptive for a poorly defined health condition and for a group of individuals which were not properly characterized. Therefore, reported elemental contents and immunohistochemical data for MMP-9, SOD, NOS are merely descriptive becoming trivial as their combined associations were not done having into account lesion types or other relevant characterization of atheroma severity. Consequently, discussion became speculative. Apart from the disregarded features mentioned above, the paper has other major inconsistencies.

We understand that the Reviewer points out that the study group is poorly defined. As we wrote in the discussion, we would be interested in any data on the history of the disease and clinical condition. We are aware that this is a serious limitation of work and we would like to change it in our project, but unfortunately it is not possible for legal reasons. Since the samples come from people who died of sudden death and are under investigation as to the cause of death, all data of the patient are confidential and cannot be viewed for scientific purposes. In Poland, where the study was conducted, there is no other option to obtain samples from deceased people.

Major concerns and study limitations:

  1. Sample characterization is missing according to the AHA lesion classification

The description of atherosclerosis grades according to AHA was supplemented in the manuscript

  1. Group characterization – how samples were distributed by lesion type and gender.

A plot of the distribution of samples in terms of sex, age and degree of atherosclerosis is provided (Fig. 1.)

  1. “Sudden death” is not defined. It can be anything from an accident to a disease condition. Criteria used for sample selection is unknown.

The cause of death was very different in each case. The cause of death was also unknown to the researchers, as it was covered by the secrets related to the conduct of a judicial investigation. Part of the deaths was definitely not related to cardiovascular diseases, and here we were hoping to obtain a "control" group without atherosclerosis, and some people may have died due to cardiovascular diseases, providing high-atherosclerotic samples. Some of the high-atherosclerotic samples could also come from accidents or murders, from the point of view of the project's assumptions, it does not seem to matter. The criterion to be followed by the forensic doctor was that the sample was of adequate quality, from people who died not earlier than 48 hours before the examination.

  1. How samples from thoracic and abdominal aorta were considered for analysis (elemental concentrations and immunohistochem) – i) two aorta regions analysed for each individual; ii) a piece of each region taken from each individual and pooled; iii) random selection.

We cut out 4 fragments from each aorta, trying to make each fragment with a different type of plaque if possible. We took samples at random for the study.

  1. Which samples and how many of each group were analysed for both elemental concentrations and immunohistochemistry

We analyzed all the samples declared in the materials and methods (from 26 pateints) in terms of the content of elements. We analyzed 24-48 sections from the same patients using the immunohistochemistry and van Kossa method (sometimes we analyzed more than 1 section from one patient. Here, the differences in the number of preparations resulted from the fact that we conducted research until we could draw conclusions). The grades of atherosclerosis for samples on IHC and ICP-OES were analogous.

  1. Elemental concentration values reported for artery vessels are extremely high, assuming that they are expressed in dry weight (dw) basis: 500-8000 mg/kg for Fe, 250-3500 mg/kg for Zn, 100-1000 mg/kg for Mn, 50-700 mg/kg for Cu, etc..

Ca concentrations reported in the paper are comparable to bone 17.5 – 33%! Even in calcified human arteries the concentration values for Ca rich regions in plaques, rarely exceed 5-6% (dw basis) (ref 1).

Unfortunately, I do not have access to the publication of Pinheiro et al. 1996. I only have access to the abstract and here I can see that other methods are used here. It is known that each method has a different sensitivity and some differences in analyte concentrations are expected when the same sample is analyzed by two different methods (this applies to all methods, e.g. protein determination with the biuret method and Bradford will also give slightly different results). As for calcium, I can only testify that some arteries were really very calcified. I can illustrate with examples of the samples we got from the forensic doctor - see photos below (photos in pdf version of answers). A large proportion of the samples had large deposits of calcium that stood out from the soft part of the tissue, and some samples were all calcified, so that attempting to straighten the tissue resulted in crushing the vessel (I also happened to break a scalpel by cutting a calcified sample).

For other elements, such as trace elements Fe and Zn, their concentrations in healthy-normal tissues, including arteries are of the order of mg/kg dw: Fe, 90-400 mg/kg (except liver and lung where Fe may reach higher levels) and Zn 40-300 mg/kg (ref 1-3). That is the reason they are called trace elements – concentrations of 10-6 and below. Average Fe and Zn contents do not seem to differ too much between non-involved tissues of the artery wall (excluding elastic lamina). However, in the atheroma Ca, Fe and Zn concentration can drastically change. In atherosclerotic human coronaries Fe and Zn decrease whereas Fe may be increased in aorta (ref 1).  Even in rabbit tissues the elemental concentrations are similar to those detected in humans (ref 4).

As mentioned above, unfortunately I do not have access to the publication of Pinheiro et al. 1996, nor Pinheiro et al. 1997, and after reading the abstract, I can only see that other methods are used here. Similar to the fourth, Minquin et al. 2003- I also have access only to the abstract and the data is based on a different technique and on rabbits. In the third study cited by Lopez et al. 2006 only measured elements in the blood, not in the artery, which also makes the results difficult to compare. It is worth noting that these works are 20-25 years old.

I understand that the Reviewer is concerned about such large discrepancies in our results and the results of other authors. We can only certify that we made every effort to ensure that the test was carried out diligently. We used one of the most sensitive measurement methods: ICP OES (Inductively Coupled Plasma Optical Emission Spectrometry) - enabling multi-element analysis of ppm (10−6) g / ml. In the case of quality control for ICP-OES data, every 10th sample is analyzed within the framework of certified standard solution containing analytical data. In this way possible drifts and sediments, etc. are controlled. Quantitative recoveries of the analytes have been investigated ensuring the data without the systematic errors.

There is no other publication that performs such analysis on the aortas with this technique. Perhaps the publication of our work will make others want to verify these results, and this would be the best point to discuss about the correctness of our data.

  1. Concentration values below detection limit were considered 0 (zero). If the value is below detection limit is a missing value in statistical terms; it cannot be assumed to be zero. Is completely wrong to establish median or average values considering missing values as 0 (zero) concentration.

The only place in our manuscript where we stated that the results below the quantification were assumed to be 0 is part of the work “3.1.1. Average analyte contents ”. We did the rest of the statistics exactly as the Reviewer writes - If the value is below detection limit is a missing value in statistical terms. As the reporting of averages without taking into account those samples where the analyte was so low that it was below the limit of quantification would greatly inflate the mean, so in section 3.1.1 we decide to completely remove the means for Cd, Cr, Pb, Mn, changing the comment.

  1. Frequency of lesion types per group is not known so any relationship established with the variable “stage of atherosclerosis” is useless. The histograms in Figures with correlations may be related to lesion type frequency classes but it is very difficult to depict anything when the population characteristics is unknown.

We hope that we have adequately described above the reasons why we cannot more precisely characterize the study group and the Reviewer finds it acceptable.

  1. Figures 5 to 8 - The immunohistochemical images are difficult to assess. No indications of artery wall stratification/tissue details, changes due to atheroma progression, etc..

We placed the main elements of the analyzed samples in the photos. We hope they're easier to interpret now.

In conclusion, authors lost the opportunity to make a relevant comparison between location of the specific proteins and lesion type with elemental concentrations. To achieve this objective, the study design should be clear and consistent. All the variables (lesion type, demographics, immunohistochem, elements) should exist for each individual. Also revising elemental concentration values produced and providing evidence of quality control for ICP-OES data is mandatory.

 Our financial capabilities and legal possibilities allowed us to carry out such a project, preventing its extension. All the variables (stage of atherosclerosis, gender, age, elements) exist for each individual. Only immunochistochemistry, as a qualitative method, does not match perfectly with the quantitative results, but it was also assumed that it was only an illustration of some dependencies, not a proof. Thanks to the Reviewer, we have many ideas for directions for the future. Regarding quality control, we hope that the Reviewer will accept our comments and give a positive opinion on our manuscript.

References

  1. Pinheiro T, et al. Nuclear microprobe applied to the study of coronary artery walls - A distinct look at atherogenesis. Cellular and Molecular Biology. 1996;42:89-102. PMID: 8833670. (complementary data in: Pallon J et al. A view on elemental distribution alterations of coronary artery walls in. Nuclear Instruments & Methods in Physics Research B. 1995;104:344-350. DOI:10.1016/0168-583X(95)00453-X.)
  2. Pinheiro T, et al. Elemental distribution in the human respiratory system and excretion organs: Absorption and accumulation. X-Ray Spectrometry. 1997;26:217-222. DOI:10.1002/(SICI)1097-4539(199707)26:4<217::AID-XRS215>3.0.CO;2-W.
  3. Lopes PA, et al. Systemic markers of the redox balance and apolipoprotein E polymorphism in atherosclerosis - The relevance for an integrated study. Biological Trace Element Research. 2006;112:57-75. DOI:10.1385/BTER:112:1:57.
  4. Minquin R, et al. Correlation of iron and zinc levels with lesion depth in newly formed atherosclerotic lesions. Free Radical Biology & Medicine, Vol. 34, No. 6, pp. 746–752, 2003. doi:10.1016/S0891-5849(02)01427-2 (complementary studies: Free Radical Biology & Medicine 38 (2005) 1206– 1211. doi:10.1016/j.freeradbiomed.2005.01.008; Free Radical Biology & Medicine 41 (2006) 222-225 doi:10.1016/j.freeradbiomed.2006.03.017)

Reviewer 2 Report

This paper showed some interesting findings but parts of the conclusion based on their observation was not solid. Furthermore, the experiments are simply without any significant mechanistic insight, although authors tried to provide some hypothesis about the mechanism involved. I will lay out some of my concerns with the work below.

Major points:

1) The authors used the Inductively Coupled Plasma - Optical Emission Spectrometer (ICP-16 OES) to determine the content of 9 ions in the aorta sections and tried to investigate the correlation between ion concentration and age/severity of disease. However, without a control group including adjacent minimally diseased regions from the same patients, it's difficult to determine those ions are specifically accumulated in the atherosclerotic lesion area or just evenly distributed along the aorta. If the contents of ions are similar among lesion areas and adjacent minimally diseased regions, it suggests the ion contents are not so closely related to atherosclerosis and may weaken the novelty of this paper which focus on aorta inomics in the context of atherosclerosis. In addition, the authors only found a negative correlations between magnesium and the severity of atherosclerosis. However, there may also exist potential correlations among other ions if comparisons are made with samples from lesion area. Therefore, it would be better to separate the human samples into maximally diseased atherosclerotic regions and adjacent minimally diseased regions for further comparison.

2) As mentioned by the authors "Four of the analytes had such low tissue contents that only some samples had detectable amounts" in Result section, the data of Table 3 is not so reliable since it was collected from limited number of samples. It would be good if the data and discussion about those four ions would be removed from this manuscript.

3) The Introduction and Discussion sections need to be re-organized. The authors tried to introduce and discuss every ion separately, but it seems better for understanding if they can re-organize those ions into groups according to their functions and correlations during atherogenesis based on previous study.

4) The authors tried to provide some significant mechanistic insights through discussion, but they didn't make any experiments to support their hypothesis. It would be appreciated if a few experiments could be added to detect the mechanism regulating atherogenesis through ion contents.

Author Response

We would like to thank the Reviewer for his many valuable comments and suggestions on how we can improve the manuscript. We tried as best we could to implement the suggestions into the manuscript and the responses to the comments are presented below (in green).

This paper showed some interesting findings but parts of the conclusion based on their observation was not solid. Furthermore, the experiments are simply without any significant mechanistic insight, although authors tried to provide some hypothesis about the mechanism involved. I will lay out some of my concerns with the work below.

Major points:

  • The authors used the Inductively Coupled Plasma - Optical Emission Spectrometer (ICP-16 OES) to determine the content of 9 ions in the aorta sections and tried to investigate the correlation between ion concentration and age/severity of disease. However, without a control group including adjacent minimally diseased regions from the same patients, it's difficult to determine those ions are specifically accumulated in the atherosclerotic lesion area or just evenly distributed along the aorta. If the contents of ions are similar among lesion areas and adjacent minimally diseased regions, it suggests the ion contents are not so closely related to atherosclerosis and may weaken the novelty of this paper which focus on aorta inomics in the context of atherosclerosis. In addition, the authors only found a negative correlations between magnesium and the severity of atherosclerosis. However, there may also exist potential correlations among other ions if comparisons are made with samples from lesion area. Therefore, it would be better to separate the human samples into maximally diseased atherosclerotic regions and adjacent minimally diseased regions for further comparison.

A valid remark is that it would be worthwhile to perform a paired sample analysis in order to analyze a plaque sample from each patient and a sample from the area without visible atherosclerosis. Unfortunately, at the stage of designing the work, we assumed that we rather wanted to analyze as many patients as possible. Due to financial reasons, we cannot currently perform a duplicate study for each patient analyzed, but we managed to perform a pilot study for three patients, where the samples were actually paired. Without statistical analysis, it was possible to conclude that in each of the three cases with low atherosclerosis, the sample contained a lot of Cu, and at high stage of atherosclerosis, little Cu. Similarly with Mg and Zn. We performed the statistical analysis with the Wilcoxon test. Unfortunately, the probably small number of paired samples meant that the results were not statistically significant. We present these results to the Reviewer here (Table I and II), but we chose not to include them in the manuscript. The study of paired samples will therefore be our direction for future research.

Table I. Content of analytes in paired samples (LLOQ: lower limit of quantification)

patient number

 Ca [mg/g]

Cd [mg/g]

Cr [mg/g]

Cu [mg/g ]

Fe [mg/g]

Mg [mg/g]

Mn [mg/g]

Pb [mg/g]

Zn [mg/g]

degree of atherosclerosis

7_11

542.62

LLOQ

LLOQ

0.14

20.36

22.45

4.00

LLOQ

1.43

2

7_7

299.91

LLOQ

0.00

0.00

0.07

5.50

0.04

0.00

0.20

6

10_11

90.00

0.06

LLOQ

0.07

2.01

6.98

0.05

LLOQ

1.02

2

10_7

322.15

0.01

LLOQ

0.02

0.39

3.91

0.05

LLOQ

0.26

6

16_12

186.14

0.05

LLOQ

0.13

0.90

12.41

0.03

LLOQ

0.69

2

16_8

231.18

LLOQ

LLOQ

0.07

37.24

5.01

LLOQ

LLOQ

0.57

6

Table II. Paired samples comparison (Wilcoxon test).

A pair of variables

Element

N

T

Z

p

atherosclerosis and control

Ca

3

3.00

0.00

1.000000

atherosclerosis and control

Cu

3

0.00

1.60

0.108810

atherosclerosis and control

Fe

3

3.00

0.00

1.000000

atherosclerosis and control

Mg

3

0.00

1.60

0.108810

atherosclerosis and control

Zn

3

0.00

1.60

0.108810

The second very interesting suggestion of the Reviewer was to separate samples only with clear atherosclerosis and to perform a correlation analysis in the hope that it will show more dependencies. A similar analysis could be performed for the control samples. We have conducted such an analysis. For the group of "high atherosclerotic" samples, only samples with atherosclerosis grade IV-VI according to AHA were taken. For the group of "control" samples only those at stages I and II were selected. This analysis, especially the one for high atherosclerotic samples, actually allowed for the emergence of a large number of dependencies. We decided to include it in the manuscript („3.2.4. Correlation analysis among high atherosclerotic and control samples”). We thank the Reviewer for this remark on how to improve the manuscript.

  • As mentioned by the authors "Four of the analytes had such low tissue contents that only some samples had detectable amounts" in Result section, the data of Table 3 is not so reliable since it was collected from limited number of samples. It would be good if the data and discussion about those four ions would be removed from this manuscript.

We understand the Reviewer's doubts. Nevertheless, we would like not to delete the results for these four parameters. We believe that the fact that the amount of these elements in most samples is so small that it is undetectable despite everything is important information. We would like to refer to other publications that also present data with the fact that the results may be below the "method limit of detection", for example Komarova et al., 2021, Liu et al., 2014.

Komarova, T.; McKeating, D.; Perkins, A.V.; Tinggi, U. Trace. Element Analysis in Whole Blood and Plasma for Reference Levels in a Selected Queensland Population, Australia. Int. J. Environ. Res. Public Health 2021, 18, 2652. https://doi.org/10.3390/ijerph18052652

Liu X, Piao J, Huang Z, et al. Determination of 16 selected trace elements in children plasma from china economical developed rural areas using high resolution magnetic sector inductively coupled mass spectrometry. J Anal Methods Chem. 2014;2014:975820. doi:10.1155/2014/975820

We marked this information clearly in the results and discussions and did not include the "problematic" results in the main tables (tab 1 and 2), only separated in Table 3, so we hope we treated this data carefully enough.

  • The Introduction and Discussion sections need to be re-organized. The authors tried to introduce and discuss every ion separately, but it seems better for understanding if they can re-organize those ions into groups according to their functions and correlations during atherogenesis based on previous study.

We analyzed the introduction and discussion again and it seems to us that the elements are basically grouped - we write separately about calcium and magnesium in the context of calcification, we also grouped elements related to antioxidant protection enzymes, we grouped heavy metals which are xenobiotics separately. If the Reviewer has a suggestion on how to combine these topics more, without causing a chaos, we will ask for advice on how to do it.

4) The authors tried to provide some significant mechanistic insights through discussion, but they didn't make any experiments to support their hypothesis. It would be appreciated if a few experiments could be added to detect the mechanism regulating atherogenesis through ion contents.

Unfortunately, this suggestion is very difficult to implement. All the experiments that we come up with in this context would have to be carried out on a different material - from living people or from animals that would be subjected to certain conditions, e.g. a specific diet / supplementation or the content of certain elements in the air. On the material from deceased people, we find no way to perform experiments that could more precisely show cause-and-effect relationships.

Once again, we would like to thank the Reviewer for all suggestions. We hope that our responses are satisfactory for the Reviewer.

Round 2

Reviewer 1 Report

As stated before my main concern remains: the paper is merely descriptive. I am afraid that the study does not bring renewed and/or new information about atherosclerosis.  

However, in general, I accept authors justifications and changes to the manuscript and I leave to the Editor’s decision whether the paper should be published or not.

Some general remarks:

There are several studies measuring aorta elemental concentrations for more than 15-20 years. Today, 5 –years old studies seem to be obsolete. I do not have any doubts that people doing chemical analysis many decades ago, some of them developed methods, validate them and proved their traceability were as skilled as today’s researchers. Instrumentation available improved, but reliability of “old” data carried out with “old instruments” are as excellent as data obtained in more modern facilities.

If in the 80’s and 90’s people were concerned about the “reference men”, nowadays aortic valves characterization became more relevant, because that information is important for clinical decision. I could send you a long list of papers using AAS, INAA, PIXE, ICP (AES, OES, MS),… to analyse aorta and other human tissues. Some of the techniques are not anymore used in analytical chemistry, such as INAA. I think searching and studying is part of your role as researchers. There were quite active groups in your country, providing the community with valuable data on human tissues.

Anyway, for future use I leave you a few other references from far away in time to more recent days:

Experimental pathology, 1986, 29(2), 119-125. https://doi.org/10.1016/S0232-1513(86)80044-5

Histochemistry 83, 87–92 (1985). https://doi.org/10.1007/BF00495306

Journal of Trace Elements in Experimental Medicine, 1991, 4(3), 173-182

X-ray Spectrom (2004), https://doi.org/10.1002/xrs.704

Exp Ther Med 7: 23-26, 2014. https://doi.org/10.3892/etm.2013.1385

J Heart Valve Dis. 2014 May;23(3):259-70. PMID: 25296447.

  1. Some specific remarks:

    1. English should be revised throughout the paper. Check phrases, verb consistency, typing errors; eg., ln: 125, 201, 263, 265, 562, 596

    1. In methods, section 2.1,

    Authors should clearly mention what was the purpose of collecting biological material

    Authors should clearly mention that aorta samples were randomly taken from thoracic and abdominal aorta territories.

    Authors should clearly mention for what you used AHA classification.

    1. 2 consider formatting x-axis: remove “sex” and identify boxes as “men” and “women”.

    Sex should be used instead of gender (e.g., ln 232.)

    1. Results, Section 3.22 is in fact methodology. Anyway text can be improved: the procedure of coding variables is usually referred as data stratification (or if you prefer separate data in two categories). In the present case this means that you want to stratify individuals (women) according to Cr, Cd and Pb concentrations (above and below DL).
    2. Results, Section 3.2.4. -  at least text has to be revised.
    3. Authors did not had controls. There are some individuals classified in AHA stages I, II which correspond to early stages or insipient atheroma development stages. Please change text accordingly.

     “High atherosclerosis/atherosclerotic” is an awkward term. Please consider to use “advanced disease stages”

    If you want to combine/emphasize the disease stage you may use, “advanced disease stages”, “large” and/or “complex (atherosclerotic) lesions”, vs “early stages” of atherosclerosis development.

    1. Discussion, ln 423, 530,.

      I would recommend to be more cautious when referring the correlation between atherosclerosis and age or that age can be a risk factor for atherosclerosis. Actually there are a consensus among those engaged in patient’s treatment and atherosclerosis research, that atheroma progresses and plaque may build up in the arteries as we age. But the association of the disease severity (progression of atheroma to critical atherosclerosis stages) with age is not straightforward and is far from being clarified. This includes the various territories exhibiting atherosclerosis: aorta, carotid, coronaries, lower limbs,… including valve problems.

    1. Copper, iron, etc “ions” are referred recurrently in the paper. As far as indicated in M&M section ICP-OES was used to measure elemental concentrations. Therefore, atoms were and measured. Not ions.

Author Response

*We appreciate the opportunity to improve the manuscript. Below are the point-by-point answers to Reviewer's comments (in green).

As stated before my main concern remains: the paper is merely descriptive. I am afraid that the study does not bring renewed and/or new information about atherosclerosis.  

*From the very beginning, this article does not comprehensively describe the etiology of atherosclerosis in the context of selected elements. However, we believe that it will bring something new to the literature of the subject. It is not one of hundreds of articles that describe such analyzes, but it is unique in terms of the number of analytes, the material analyzed and also the number of samples (compared to other works) is not small.

However, in general, I accept authors justifications and changes to the manuscript and I leave to the Editor’s decision whether the paper should be published or not.

Some general remarks:

There are several studies measuring aorta elemental concentrations for more than 15-20 years. Today, 5 –years old studies seem to be obsolete. I do not have any doubts that people doing chemical analysis many decades ago, some of them developed methods, validate them and proved their traceability were as skilled as today’s researchers. Instrumentation available improved, but reliability of “old” data carried out with “old instruments” are as excellent as data obtained in more modern facilities.

If in the 80’s and 90’s people were concerned about the “reference men”, nowadays aortic valves characterization became more relevant, because that information is important for clinical decision. I could send you a long list of papers using AAS, INAA, PIXE, ICP (AES, OES, MS),… to analyse aorta and other human tissues. Some of the techniques are not anymore used in analytical chemistry, such as INAA. I think searching and studying is part of your role as researchers. There were quite active groups in your country, providing the community with valuable data on human tissues.

Anyway, for future use I leave you a few other references from far away in time to more recent days:

Experimental pathology, 1986, 29(2), 119-125. https://doi.org/10.1016/S0232-1513(86)80044-5

Histochemistry 83, 87–92 (1985). https://doi.org/10.1007/BF00495306

Journal of Trace Elements in Experimental Medicine, 1991, 4(3), 173-182 ZACYTOWANO Dubick ET AL

X-ray Spectrom (2004), https://doi.org/10.1002/xrs.704 ZACYTOWANO WRÓBEL ET AL

Exp Ther Med 7: 23-26, 2014. https://doi.org/10.3892/etm.2013.1385 Jin Hyun Joh

J Heart Valve Dis. 2014 May;23(3):259-70. PMID: 25296447.

*We do not discredit older works. However, we are taught that we should reach for works from the last 10 years. Typically, Reviewers accused us of using older papers. In previous replies to reviews our comment was added just to explain where the differences in values may be.

*Thank you for the literature proposal. I acquainted with all the proposed publications. Those from 1995 and 86- unfortunately only with abstracts, because the rest is unavailable. The works by Dubicks, Wróbel and Joh seemed so interesting that we decided to quote them. 

  1. Some specific remarks:
    1. English should be revised throughout the paper. Check phrases, verb consistency, typing errors; eg., ln: 125, 201, 263, 265, 562, 596

*It is difficult not to overlook some mistakes in such a long text. We apologize for them. We reread the work and corrected the errors we noticed.

*Unfortunately, we could not use the mentioned line numbers, because in our version the lines were clearly "shifted", even in the pdf version, which we sent after the first corrections, the lines do not match, eg in line 263 there is no text at all.

    1. In methods, section 2.1,

Authors should clearly mention what was the purpose of collecting biological material

*It was written in the first sentence of paragraph 2.1, I quote "which were part of the biological material taken for histopathological examination to determine the cause of death of the deceased." In our opinion, this is a sufficient explanation. If the reviewer would like any specific information added, please indicate what information is missing.

Authors should clearly mention that aorta samples were randomly taken from thoracic and abdominal aorta territories.

*Added this statement to the manuscript

Authors should clearly mention for what you used AHA classification.

*Added the statement to the manuscript 

    1. 2 consider formatting x-axis: remove “sex” and identify boxes as “men” and “women”.

*Corrected, actually much more readable.

Sex should be used instead of gender (e.g., ln 232.)

*Corrected.

    1. Results, Section 3.22 is in fact methodology. Anyway text can be improved: the procedure of coding variables is usually referred as data stratification (or if you prefer separate data in two categories). In the present case this means that you want to stratify individuals (women) according to Cr, Cd and Pb concentrations (above and below DL).

*Modified

    1. Results, Section 3.2.4. -  at least text has to be revised.

*Revised

    1. Authors did not had controls. There are some individuals classified in AHA stages I, II which correspond to early stages or insipient atheroma development stages. Please change text accordingly.

*The entry "controls" has been deleted and the wording has been changed accordingly.

 “High atherosclerosis/atherosclerotic” is an awkward term. Please consider to use “advanced disease stages”

If you want to combine/emphasize the disease stage you may use, “advanced disease stages”, “large” and/or “complex (atherosclerotic) lesions”, vs “early stages” of atherosclerosis development.

*The wording has been changed.

    1. Discussion, ln 423, 530,.

  I would recommend to be more cautious when referring the correlation between atherosclerosis and age or that age can be a risk factor for atherosclerosis. Actually there are a consensus among those engaged in patient’s treatment and atherosclerosis research, that atheroma progresses and plaque may build up in the arteries as we age. But the association of the disease severity (progression of atheroma to critical atherosclerosis stages) with age is not straightforward and is far from being clarified. This includes the various territories exhibiting atherosclerosis: aorta, carotid, coronaries, lower limbs,… including valve problems.

*Indeed, the relationship between the development of atherosclerosis and age is ambiguous, and the correlation between such data is certainly not close 1. We tried to soften our statement so that there would be no mistaken impression that we thought otherwise.

    1. Copper, iron, etc “ions” are referred recurrently in the paper. As far as indicated in M&M section ICP-OES was used to measure elemental concentrations. Therefore, atoms were and measured. Not ions.

*We modified any fragments that might suggest that we were analyzing the ions. We decided to change the title as well, because in the original version the word "ionomics" was used. Thank you for this apt remark. 

*In conclusion, we thank to the Reviewer for the contribution that he made in supplementing and revising this manuscript. We appreciate the Reviewer's experience in the field of the manuscript and are grateful for all opinions and suggestions. We hope that this version will be satisfactory for the Reviewer and will receive a positive opinion to the Biomolecules editorial office.

Reviewer 2 Report

The revised manuscript presented improved quality after adding control group and figure editing, although the small amount of samples from control group made the statistics almost useless.

There are still some small errors in the text needed to be fixed up. For example, the authors state "No statistically significant relationship was observed for women. For men, there was a correlation between Ca-age, Cu-age, Fe-age, Mg- degree of atherosclerosis. The r values are in Table 4, and the scatter plots in Fig. 5". However, in the table legend of Tab. 4, the authors describe the table content as " The values of the correlation coefficient for the content of analytes in the tissue of women considering age and stage of atherosclerosis. Statistically significant values are marked with an asterisk". It looks like the "women" here is mislabeled and should be "men".

Author Response

*Thank you for your favor. Below is our answer to the second round of the review (in green).

The revised manuscript presented improved quality after adding control group and figure editing, although the small amount of samples from control group made the statistics almost useless.

There are still some small errors in the text needed to be fixed up. For example, the authors state "No statistically significant relationship was observed for women. For men, there was a correlation between Ca-age, Cu-age, Fe-age, Mg- degree of atherosclerosis. The r values are in Table 4, and the scatter plots in Fig. 5". However, in the table legend of Tab. 4, the authors describe the table content as " The values of the correlation coefficient for the content of analytes in the tissue of women considering age and stage of atherosclerosis. Statistically significant values are marked with an asterisk". It looks like the "women" here is mislabeled and should be "men".

*Thank you very much for this rightful remark regarding the incorrect caption under Table 4. We have corrected this error and several others noticed by us or by the other Reviewer. We hope you find this version of the manuscript flawless and worth publishing in Biomolecules. Thanks again for your time and attention to our manuscript.